# A Maximum Mutual Information Framework for Multi-Agent Reinforcement Learning

## Abstract

In this paper, we propose a maximum mutual information (MMI) framework for multi-agent reinforcement learning (MARL) to enable multiple agents to learn coordinated behaviors by regularizing the accumulated return with the mutual information between actions. By introducing a latent variable to induce nonzero mutual information between actions and applying a variational bound, we derive a tractable lower bound on the considered MMI-regularized objective function. Applying policy iteration to maximize the derived lower bound, we propose a practical algorithm named variational maximum mutual information multi-agent actor-critic (VM3-AC), which follows centralized learning with decentralized execution (CTDE). We evaluated VM3-AC for several games requiring coordination, and numerical results show that VM3-AC outperforms other MARL algorithms in multi-agent tasks requiring coordination.

## 1 Introduction

With the success of RL in the single-agent domain (Mnih et al. (2015); Lillicrap et al. (2015)), MARL is being actively studied and applied to real-world problems such as traffic control systems and connected self-driving cars, which can be modeled as multi-agent systems requiring coordinated control (Li et al. (2019); Andriotis & Papakonstantinou (2019)). The simplest approach to MARL is independent learning, which trains each agent independently while treating other agents as a part of the environment. One such example is independent Q-learning (IQL) (Tan (1993)), which is an extension of Q-learning to multi-agent setting. However, this approach suffers from the problem of non-stationarity of the environment. A common solution to this problem is to use fully-centralized critic in the framework of centralized training with decentralized execution (CTDE) (OroojlooyJadid & Hajinezhad (2019); Rashid et al. (2018)). For example, MADDPG (Lowe et al. (2017)) uses a centralized critic to train a decentralized policy for each agent, and COMA (Foerster et al. (2018)) uses a common centralized critic to train all decentralized policies. However, these approaches assume that decentralized policies are independent and hence the joint policy is the product of each agent's policy. Such non-correlated factorization of the joint policy limits the agents to learn coordinated behavior due to negligence of the influence of other agents (Wen et al. (2019); de Witt et al. (2019)). Thus, learning coordinated behavior is one of the fundamental problems in MARL (Wen et al. (2019); Liu et al. (2020)).

In this paper, we introduce a new framework for MARL to learn coordinated behavior under CTDE. Our framework is based on regularizing the expected cumulative reward with mutual information among agents' actions induced by injecting a latent variable. The intuition behind the proposed framework is that agents can coordinate with other agents if they know what other agents will do with high probability, and the dependence between action policies can be captured by the mutual information. High mutual information among actions means low uncertainty of other agents' actions. Hence, by regularizing the objective of the expected cumulative reward with mutual information among agents' actions, we can coordinate the behaviors of agents implicitly without explicit dependence enforcement. However, the optimization problem with the proposed objective function has several difficulties since we consider decentralized policies without explicit dependence or communication in the execution phase. In addition, optimizing mutual information is difficult because of the intractable conditional distribution. We circumvent these difficulties by exploiting the property of the latent variable injected to induce mutual information, and applying variational lower bound on the mutual

information. With the proposed framework, we apply policy iteration by redefining value functions to propose the VM3-AC algorithm for MARL with coordinated behavior under CTDE.

## 2 RELATED WORK

Learning coordinated behavior in multi-agent systems is studied extensively in the MARL community. To promote coordination, some previous works used communication among agents (Zhang & Lesser (2013); Foerster et al. (2016); Pesce & Montana (2019)). For example, Foerster et al. (2016) proposed the DIAL algorithm to learn a communication protocol that enables the agents to coordinate their behaviors. Instead of relying on communication, Jaques et al. (2018) proposed social influence intrinsic reward which is related to the mutual information between actions to achieve coordination. The purpose of the social influence approach is similar to our approach and the social influence yields good performance in social dilemma environments. The difference between our algorithm and the social influence approach will be explained in detail and the effectiveness of our approach over the social influence approach will be shown in Section 6. Wang et al. (2019) proposed an intrinsic reward capturing the influence based on the mutual information between an agent's current actions/states and other agents' next states. In addition, they proposed an intrinsic reward based on a decision-theoretic measure. Although they used the mutual information to enhance exploration, our approach focuses on the mutual information between simultaneous actions capturing policy correlation not influence. Besides, they considered independent policies, whereas policies are correlated in our approach.

Some previous works considered correlated policies instead of independent policies. For example, Liu et al. (2020) proposed explicit modeling of correlated policies for multi-agent imitation learning, and Wen et al. (2019) proposed a recursive reasoning framework for MARL to maximize the expected return by decomposing the joint policy into own policy and opponents' policies. Going beyond adopting correlated policies, our approach maximizes the mutual information between actions which is a measure of correlation.

Our framework can be interpreted as enhancing correlated exploration by increasing the entropy of own policy while decreasing the uncertainty about other agents' actions. Some previous works proposed other techniques to enhance correlated exploration (Zheng & Yue (2018); Mahajan et al. (2019)). For example, MAVEN addressed the poor exploration problem of QMIX by maximizing the mutual information between the latent variable and the observed trajectories (Mahajan et al. (2019)). However, MAVEN does not consider the correlation among policies.

## 3 BACKGROUND

We consider a Markov Game (Littman (1994)), which is an extention of Markov Decision Process (MDP) to multi-agent setting. An $N$-agent Markov game is defined by an environment state space $\mathcal{S}$, action spaces for $N$ agents $\mathcal{A}_1, \cdots, \mathcal{A}_N$, a state transition probability $\mathcal{T} : \mathcal{S} \times \mathcal{A} \times \mathcal{S} \to [0, 1]$, where $\mathcal{A} = \prod_{i=1}^{N} \mathcal{A}_i$ is the joint action space, and a reward function $\mathcal{R} : \mathcal{S} \times \mathcal{A} \to \mathbb{R}$. At each time step $t$, Agent $i$ executes action $a_t^i \in \mathcal{A}_i$ based on state $s_t \in \mathcal{S}$. The actions of all agents $\boldsymbol{a}_t = (a_t^1, \cdots, a_t^N)$ yield next state $s_{t+1}$ according to $\mathcal{T}$ and yield shared common reward $r_t$ according to $\mathcal{R}$ under the assumption of fully-cooperative MARL. The discounted return is defined as $R_t = \sum_{\tau=t}^{\infty} \gamma^\tau r_\tau$, where $\gamma \in [0, 1)$ is the discounting factor.

We assume CTDE incorporating resource asymmetry between training and execution phases, widely considered in MARL (Lowe et al. (2017); Iqbal & Sha (2018); Foerster et al. (2018)). Under CTDE, each agent can access all information including the environment state, observations and actions of other agents in the training phase, whereas the policy of each agent can be conditioned only on its own action-observation history $\tau_t^i$ or observation $o_t^i$ in the execution phase. For given joint policy $\boldsymbol{\pi} = (\pi^1, \cdots, \pi^N)$, the goal of fully cooperative MARL is to find the optimal joint policy $\boldsymbol{\pi}^*$ that maximizes the objective $J(\boldsymbol{\pi}) = E_{\boldsymbol{\pi}}[R_0]$.

**Maximum Entropy RL** The goal of maximum entropy RL is to find an optimal policy that maximizes the entropy-regularized objective function, given by

$$J(\pi) = E_\pi \left[ \sum_{t=0}^{\infty} \gamma^t \Big( r_t(s_t, a_t) + \alpha H(\pi(\cdot|s_t)) \Big) \right] \tag{1}$$

It is known that this objective encourages the policy to enhance exploration in the state and action spaces and helps the policy avoid converging to a local minimum. Soft actor-critic (SAC), which is based on the maximum entropy RL principle, approximates soft policy iteration to the actor-critic method. SAC outperforms other deep RL algorithms in many continuous action tasks (Haarnoja et al. (2018)).

We can simply extend SAC to multi-agent setting in the manner of independent learning. Each agent trains its decentralized policy using decentralized critic to maximize the weighted sum of the cumulative return and the entropy of its policy. We refer to this method as Independent SAC (I-SAC). Adopting the framework of CTDE, we can replace decentralized critic with centralized critic which incorporates observations and actions of all agents. We refer to this method as multi-agent soft actor-critic (MA-SAC). Both I-SAC and MA-SAC are considered as baselines in the experiment section.

## 4 THE PROPOSED MAXIMUM MUTUAL INFORMATION FRAMEWORK

We assume that the environment is fully observable, i.e., each agent can observe the environment state $s_t$ for theoretical development in this section, and will consider partially observable environment for practical algorithm construction under CTDE in the next section.

Under the proposed MMI framework, we aims to find the policy that maximizes the mutual information between actions in addition to cumulative return. Thus, the MMI-regularized objective function for joint policy $\boldsymbol{\pi}$ is given by

$$J(\boldsymbol{\pi}) = E_{\boldsymbol{\pi}}\left[\sum_{t=0}^{\infty}\gamma^t\Big(r_t(s_t, \boldsymbol{a_t}) + \alpha\sum_{(i,j)}I(\pi^i(\cdot|s_t); \pi^j(\cdot|s_t))\Big)\right] \tag{2}$$

where $a_t^i \sim \pi^i(\cdot|s_t)$ and $\alpha$ is the temperature parameter that controls the relative importance of the mutual information against the reward.

As aforementioned, we assume decentralized policies and want the decentralized policies to exhibit coordinated behavior. By regularization with mutual information in the proposed objective function (2), the policy of each agent is implicitly encouraged to coordinate with other agents' policies without explicit dependency by reducing the uncertainty about other agents' policies. This can be seen as follows: Mutual information is expressed in terms of entropy and conditional entropy as

$$I(\pi^i(\cdot|s_t); \pi^j(\cdot|s_t)) = H(\pi^j(\cdot|s_t)) - H(\pi^j(\cdot|s_t)|\pi^i(\cdot|s_t)). \tag{3}$$

If the knowledge of $\pi^i(\cdot|s_t)$ does not provide any information about $\pi^j(\cdot|s_t)$, the conditional entropy reduces to the unconditional entropy, i.e., $H(\pi^j(\cdot|s_t)|\pi^i(\cdot|s_t)) = H(\pi^j(\cdot|s_t))$, and the mutual information becomes zero. Maximizing mutual information is equivalent to minimizing the uncertainty about other agents' policies conditioned on the agent's own policy, which can lead the agent to learn coordinated behavior based on the reduced uncertainty about other agents' policies.

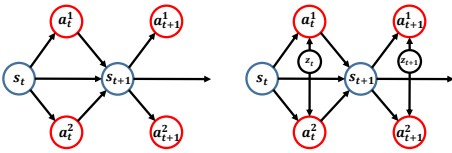

However, direct optimization of the objective function (2) is not easy. Fig. 1(a) shows the causal diagram of the considered system model described in Section 3 in the case of two agents with decentralized policies. Since we consider the case of no explicit dependency, the two policy distributions can be expressed as $\pi^1(a_t^1|s_t)$ and $\pi^2(a_t^2|s_t)$. Then, for given environment state $s_t$ observed by both agents, $\pi^1(a_t^1|s_t)$ and $\pi^2(a_t^2|s_t)$ are conditionally independent and the mutual information $I(\pi^1(\cdot|s_t); \pi^2(\cdot|s_t)) = 0$. Thus, the MMI objective

Figure 1: Causal diagram in 2-agent Markov Game: (a) Standard MARL, (b) Introducing the latent variable to the standard MARL

(2) reduces to the standard MARL objective of only the accumulated return. In the following subsections, we present our approach to circumvent this difficulty and implement the MMI framework and its operation under CTDE.

### 4.1 Inducing Mutual Information Using Latent Variable

First, in order to induce mutual information among agents' policies under the considered system causal diagram shown in Fig. 1(a), we introduce latent variable $z_t$. For illustration, consider the new diagram with latent variable $z_t$ in Fig. 1(b). Suppose that the latent variable $z_t$ has a prior distribution $p(z_t)$, and assume that both actions $a_t^1$ and $a_t^2$ are generated from the *observed* random variable $s_t$ and the *unobserved* random variable $z_t$. Then, the policy of Agent $i$ is given by the marginal distribution $\pi^i(\cdot|s_t) = \int_z \pi^i(\cdot|s_t, z)p(z)dz$ marginalized over $z$. With the unobserved latent random variable $z$, the conditional independence does not hold for $a_t^1$ and $a_t^2$ and the mutual information can be positive, i.e., $I(\pi^1(\cdot|s_t); \pi^2(\cdot|s_t)) > 0$. Hence, we can induce the mutual information between actions without explicit dependence by introducing the latent variable. In the general case of $N$ agents, we have $\boldsymbol{\pi}(a^1, \cdots, a^N|s) = E_z[\pi^1(a^1|s, z) \cdots \pi^N(a^N|s, z)]$. Note that in this case we inject a common latent variable $z$ into all agents' policies.

### 4.2 Variational Bound of Mutual Information

Even with non-trivial mutual information $I(\pi^i(\cdot|s_t); \pi^j(\cdot|s_t))$, it is difficult to directly compute the mutual information. Note that we need the conditional distribution of $a_t^j$ given $(a_t^i, s_t)$ to compute the mutual information as seen in (4), but it is difficult to know the conditional distribution directly. To circumvent this difficulty, we use a variational distribution $q(a_t^j|a_t^i, s_t)$ to approximate $p(a_t^j|a_t^i, s_t)$ and derive a lower bound on the mutual information $I(\pi^i(\cdot|s_t); \pi^j(\cdot|s_t)) =: I_{ij}(s_t)$ as

$$I_{ij}(s_t) = E_{p(a_t^i, a_t^j|s_t)}\left[\log\frac{q(a_t^j|a_t^i, s_t)}{p(a_t^j)}\right] + E_{p(a_t^i|s_t)}\left[KL(p(a_t^j|a_t^i, s_t)\|q(a_t^j|a^i, s_t))\right]$$

$$\geq H(\pi^j(\cdot|s_t)) + E_{p(a_t^i, a_t^j|s_t)}\left[\log q(a_t^j|a_t^i, s_t)\right], \tag{4}$$

where the inequality holds because KL divergence is always non-negative. The lower bound becomes tight when $q(a_t^j|a_t^i, s_t)$ approximates $p(a_t^j|a_t^i, s_t)$ well. Using the symmetry of mutual information, we can rewrite the lower bound as

$$I_{ij}(s_t) \geq \frac{1}{2}\left[H(\pi^i(\cdot|s_t)) + H(\pi^j(\cdot|s_t)) + E_{p(a_t^i, a_t^j|s_t)}\left[\log q(a_t^i|a_t^j, s_t) + \log q(a_t^j|a_t^i, s_t)\right]\right]. \tag{5}$$

Then, we can maximize this lower bound of mutual information by using the tractable approximation $q(a_t^i|a_t^j, s_t)$.

### 4.3 Modified Policy Iteration

In this subsection, we develop policy iteration for the MMI framework. First, we replace the original MMI objective function (2) with the following tractable objective function based on the variational lower bound (5):

$$\hat{J}(\boldsymbol{\pi}, q) = E_{\boldsymbol{\pi}}\left[\sum_{t=0}^{\infty}\gamma^t\Big(r_t(s_t, \boldsymbol{a_t}) + \alpha N \sum_{i=1}^{N} H(\pi^i(\cdot|s_t)) + \alpha \sum_{i=1}^{N}\sum_{j\neq i}\log q(a_t^j|a_t^i, s_t)\Big)\right], \tag{6}$$

where $q(a_t^j|a_t^i, s_t)$ is the variational distribution to approximate the conditional distribution $p(a_t^j|a_t^i, s_t)$. Then, we determine the individual objective function $\hat{J}^i(\pi^i, q)$ for Agent $i$ as the sum of the terms in (6) associated with Agent $i$'s policy $\pi^i$ or action $a_t^i$, given by $\hat{J}^i(\pi^i, q) =$

$$E_{\boldsymbol{\pi}}\left[\sum_{t=0}^{\infty}\gamma^t\Big(\underbrace{r_t(s_t, \boldsymbol{a_t}) + \beta \cdot H(\pi^i(\cdot|s_t))}_{(a)} + \frac{\beta}{N}\sum_{j\neq i}\Big[\underbrace{\log q(a_t^i|a_t^j, s_t) + \log q(a_t^j|a_t^i, s_t)}_{(b)}\Big]\Big)\right], \tag{7}$$

where $\beta = \alpha N$ is the temperature parameter. Note that maximizing the term (a) in (7) implies that each agent maximizes the weighted sum of the policy entropy and the return, which can be interpreted as an extension of maximum entropy RL to multi-agent setting. On the other hand, maximizing the term (b) with respect to $\pi^i$ means that we update the policy $\pi^i$ so that Agent $j$ well predicts Agent $i$'s action by the first term in (b) and Agent $i$ well predicts Agent $j$'s action by the

second term in (b). Thus, the objective function (7) can be interpreted as the maximum entropy MARL objective combined with predictability enhancement for other agents' actions. Note that predictability is reduced when actions are uncorrelated. Since the policy entropy term $H(\pi^i(\cdot|s_i))$ enhances individual exploration due to maximum entropy principle (Haarnoja et al. (2018)) and the term (b) in (7) enhances predictability or correlation among agents' actions, the proposed objective function (7) can be considered as one implementation of the concept of *correlated exploration* in MARL (Mahajan et al. (2019)).

Now, in order to learn policy $\pi^i$ to maximize the objective function (7), we modify the policy iteration in standard RL. For this, we redefine the state and state-action value functions for each agent as follows:

$$V_i^{\boldsymbol{\pi}}(s) \triangleq E_{\boldsymbol{\pi}}\left[\sum_{t=0}^{\infty} \gamma^t \left(r_t + \beta H(\pi^i(\cdot|s_t)) + \frac{\beta}{N}\sum_{j \neq i} \log q^{(i,j)}(a_t^i, a_t^j, s_t)\right)\bigg| s_0 = s\right] \quad (8)$$

$$Q_i^{\boldsymbol{\pi}}(s,a) \triangleq E_{\boldsymbol{\pi}}\left[r_0 + \gamma V_i^{\boldsymbol{\pi}}(s_1)\bigg| s_0 = s, a_0 = a\right], \quad (9)$$

where $q^{(i,j)}(a_t^i, a_t^j, s_t) \triangleq q(a_t^i|a_t^j, s_t)q(a_t^j|a_t^i, s_t)$. Then, the Bellman operator corresponding to $V_i^{\boldsymbol{\pi}}$ and $Q_i^{\boldsymbol{\pi}}$ is given by

$$\mathcal{T}^{\boldsymbol{\pi}}Q_i(s,\boldsymbol{a}) \triangleq r(s,\boldsymbol{a}) + \gamma E_{s' \sim p}[V_i(s')], \quad (10)$$

where

$$V_i(s) = E_{\boldsymbol{a} \sim \boldsymbol{\pi}}\left[Q_i(s,\boldsymbol{a}) - \beta \log \pi^i(a^i|s) + \frac{\beta}{N}\sum_{j \neq i} \log q^{(i,j)}(a^i, a^j, s)\right] \quad (11)$$

In the policy evaluation step, we compute the value functions defined in (14) and (15) by applying the modified Bellman operator $\mathcal{T}^{\boldsymbol{\pi}}$ repeatedly to any initial function $Q_i^0$.

**Lemma 1.** *(Variational Policy Evaluation). For fixed $\boldsymbol{\pi}$ and the variational distribution $q$, consider the modified Bellman operator $\mathcal{T}^{\boldsymbol{\pi}}$ in (16) and an arbitrary initial function $Q_i^0 : \mathcal{S} \times \mathcal{A} \to \mathbb{R}$, and define $Q_i^{k+1} = \mathcal{T}^{\boldsymbol{\pi}}Q_i^k$. Then, $Q_i^k$ converges to $Q_i^{\boldsymbol{\pi}}$ defined in (15).*

*Proof.* See Appendix A.

In the policy improvement step, we update the policy and the variational distribution by using the value function evaluated in the policy evaluation step. Here, each agent updates its policy and variational distribution while keeping other agents' policies fixed as follows: $(\pi_{k+1}^i, q_{k+1}) =$

$$\arg\max_{\pi^i, q} E_{(a^i, a^{-i}) \sim (\pi^i, \pi_k^{-i})}\left[Q_i^{\boldsymbol{\pi}_k}(s,\boldsymbol{a}) - \beta \log \pi^i(a^i|s) + \frac{\beta}{N}\sum_{j \neq i} \log q^{(i,j)}(a^i, a^j, s))\right], \quad (12)$$

where $a^{-i} \triangleq \{a^1, \cdots, a^N\} \backslash \{a^i\}$. Then, we have the following lemma regarding the improvement step.

**Lemma 2.** *(Variational Policy Improvement). Let $\pi_{new}^i$ and $q_{new}$ be the updated policy and the variational distribution from (30) in Appendix A. Then, $Q_i^{\pi_{new}^i, \pi_{old}^{-i}}(s,\boldsymbol{a}) \geq Q_i^{\pi_{old}^i, \pi_{old}^{-i}}(s,\boldsymbol{a})$ for all $(s,\boldsymbol{a}) \in (\mathcal{S} \times \boldsymbol{\mathcal{A}})$.*

*Proof.* See Appendix A.

The modified policy iteration is defined as applying the variational policy evaluation and variational improvement steps in an alternating manner. Each agent trains its policy, critic and the variational distribution to maximize its objective function (7).

## 5 ALGORITHM CONSTRUCTION

Summarizing the development above, we now propose the variational maximum mutual information multi-agent actor-critic (VM3-AC) algorithm, which can be applied to continuous and partially

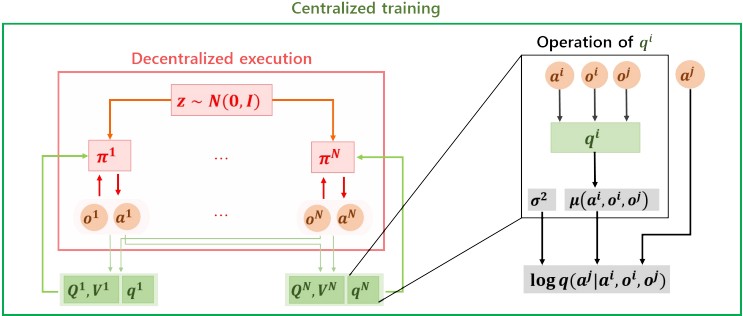

Figure 2: Overall operation of the proposed VM3-AC. We only need the operation in the red box after training.

observable multi-agent environments under CTDE. The overall operation of VM3-AC is shown in Fig. 2. Under CTDE, each agent's policy is conditioned only on local observation, and centralized critics are conditioned on either the environment state or the observations of all agents, depending on the situation (Lowe et al. (2017)). Let $\boldsymbol{x}$ denote either the environment state $s$ or the observations of all agents $(o_1, \cdots, o_N)$, whichever is used. In order to deal with the large continuous state-action spaces, we adopt deep neural networks to approximate the required functions. For Agent $i$, we parameterize the variational distribution with $\xi^i$ as $q_{\xi^i}(a^j|a^i, o^i, o^j)$, the state-value function with $\psi^i$ as $V_{\psi_i}^i(\boldsymbol{x})$, two action-value functions with $\theta^{i,1}$ and $\theta^{i,2}$ as $Q_{\theta^{i,1}}^i(\boldsymbol{x}, \boldsymbol{a}), Q_{\theta^{i,2}}^i(\boldsymbol{x}, \boldsymbol{a})$, and the policy with $\phi^i$ as $\pi_{\phi^i}^i(a|o^i) = E_z[\pi_{\phi^i}^i(a|o^i, z)]$. We assume normal distribution for the latent variable which plays a key role in inducing coordination among agents' policies, i.e., $z_t \sim \mathcal{N}(0, I)$, and further assume that the variational distribution is Gaussian distribution with constant variance $\sigma^2$, i.e., $q_{\xi^i}(a^j|a^i, o^i, o^j) = \mathcal{N}(\mu_{\xi^i}(a^i, o^i, o^j), \sigma^2)$, where $\mu_{\xi^i}(a^i, o^i, o^j)$ is the mean of the distribution.

**Centralized Training** As aforementioned, the policy is the marginalized distribution over the latent variable $z$, where the policies of all agents take the same $z_t$ generated from $\mathcal{N}(0, I)$ as an input variable. We perform the required marginalization based on Monte Carlo numerical expectation as follows:

$$\boldsymbol{\pi}(\boldsymbol{a}|s) = E_z[\pi_{\phi^1}^1(a^1|s, z) \cdots \pi_{\phi^N}^N(a^N|s, z)] \simeq \frac{1}{L}\sum_{l=1}^{L} \pi_{\phi^1}^1(a^1|s, z^l) \cdots \pi_{\phi^N}^N(a^N|s, z^l), \quad (13)$$

and we use $L = 1$ for simplicity. The parameterized value functions, policy, and variational distributions are trained similarly to the training in SAC. Due to space limitation, training detail and pseudo code are provided in Appendices B and C.

**Decentralized Execution** In the centralized training phase, we pick actions $(a^1, \cdots, a^N)$ by using Monte Carlo expectation based on common latent variable $z^l$ generated from zero-mean Gaussian distribution, as seen in (13). We consider two methods to achieve the same operation in the decentralized execution phase. First, this can be done by making all agents have the same Gaussian random sequence generator and distributing the same seed to this random sequence generator only once in the beginning of the execution phase. This eliminates the necessity of communication for sharing the latent variable. In fact, this way of sharing $z^l$ can be applied to the centralized training phase too. Second, we exploit the property of zero-mean Gaussian latent variable $z^l$. That is, we simply replace $z^l$ with zero vector with the matching dimensions in the decentralized execution phase. This substitution method does not deteriorate the performance much as seen in Section 6 since the latent variable distribution is zero-mean Gaussian and zero has the highest density. Thus, the proposed algorithm is fully operative under CTDE.

## 6 EXPERIMENT

In this section, we provide numerical results to evaluate VM3-AC. We considered four baselines: 1) MADDPG (Lowe et al. (2017)) - an extension of DDPG with a centralized critic to train a

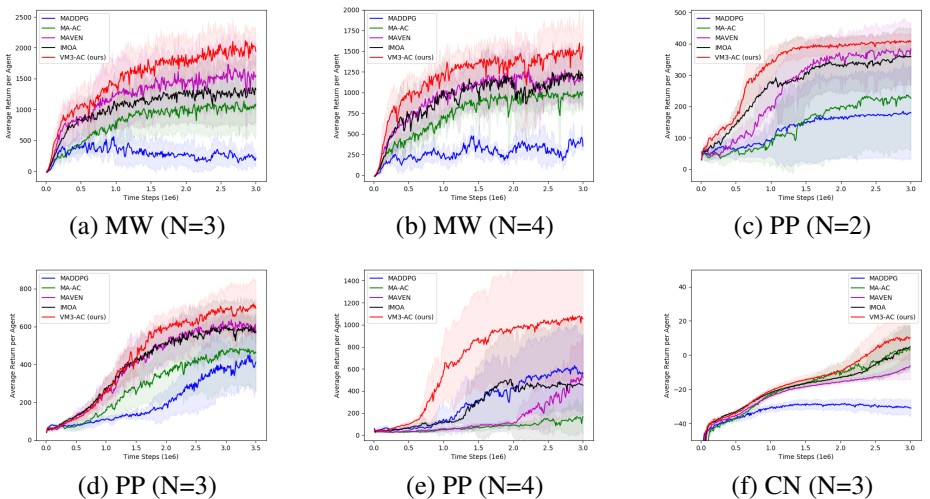

(a) MW (N=3)  (b) MW (N=4)  (c) PP (N=2)

(d) PP (N=3)  (e) PP (N=4)  (f) CN (N=3)

Figure 3: Performance of MADDPG (blue), MA-AC (green), MAVEN (purple), IMOA (black), and VM3-AC (the proposed method, red) on multi-walker environments (a)-(b), predator-prey (c)-(e), and cooperative navigation (f). (MW, PP, and CN denote multi-walker, predator-prey, and cooperative navigation, respectively)

decentralized policy for each agent. 2) Multi-agent actor-critic (MA-AC) - a variant of VM3-AC ($\beta = 0$) without the latent variable. 3) Multi-agent variational exploration (MAVEN) (Mahajan et al. (2019)). Similarly to VM3-AC, MAVEN introduced latent variable and variational approach for optimizing the mutual information. However, MAVEN does not consider the mutual information between actions but consider the mutual information between the latent variable and trajectories of the agents. 4) Influence MOA (IMOA) also known as social influence (SI) (Jaques et al. (2018)). The IMOA method models $p(a_{t+1}^j|a_t^i, s_t^i)$, where agents $j$ and $i$ are influencee and influencer, respectively, and adds intrinsic reward given by the mutual information $I(a_t^i; a_{t+1}^j|s_t)$ between influencer's current action and influencee's next timestep action **not** the mutual information $I(a_t^i; a_t^j|s_t)$ between two agents' simultaneous actions, which is considered in our MMI framework. Both MAVEN and IMOA are implemented on the top of MA-AC since we consider continuous action-space environments. We evaluated the proposed algorithm and the baselines in three multi-agent environments with the varying number of agents: multi-walker (Gupta et al. (2017)), predator-prey (Lowe et al. (2017)), and cooperative navigation (Lowe et al. (2017)). We modified the original environments to require further coordination among agents. For example, we increased the size of the agent and collision reward in the cooperative navigation. Hence, the agent should consider other agents more while achieving its goal. The detailed setting of each environments is provided in Appendix D.

## 6.1 RESULT

Fig. 3 shows the learning curves for the considered three environments with the different number of agents. The y-axis denotes the average of all agents' rewards averaged over 7 random seeds, and the x-axis denotes time step. The hyperparameters including the temperature parameter $\beta$ and the dimension of the latent variable are provided in Appendix E. As shown in Fig. 3, VM3-AC outperforms the baselines in the considered environments. Especially, in the case of the multi-walker environment, VM3-AC has large performance gain. This is because the agents in the multi-walker environment are required especially to learn coordinated behavior to obtain high rewards. In addition, the agents in the predator-prey environment, where the number of agents is four, should spread out in groups of two to get more reward. In this environment, VM3-AC also has large performance gain. Thus, it is seen that the proposed MMI framework improves performance in complex multi-agent tasks requiring high-quality coordination. It is observed that both MAVEN and IMOA outperform the basic algorithm MA-AC but not VM3-AC. Hence, the mutual information between the latent variable and trajectory (used in MAVEN) and the mutual information between the action of the agent and the

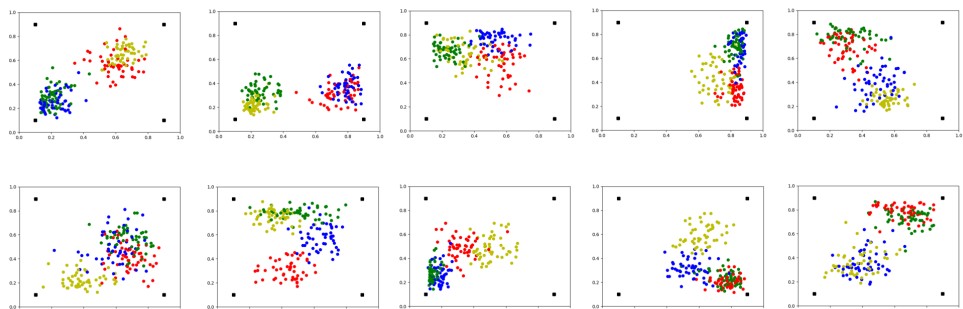

Figure 4: The positions of four agents after five time-steps after the episode begins in the early stage of the training: 1st row - VM3-AC and 2nd row - MA-SAC. The figures in column correspond to a different seed. The black squares are the preys and each color except black shows the position of each agent.

next time step action of other agents (used in IMOA) are not as effective for coordinated behavior as the mutual information between agents' simultaneous actions used for VM3-AC.

## 6.2 ABLATION STUDY

In this section, we provide ablation study on the major techniques and hyperparameter of VM3-AC: 1) mutual information versus entropy 2) the latent variable, 3) the temperature parameter $\beta$, and 4) injecting zero vector instead of the latent variable $z$ to policies in the execution phase.

**Mutual information versus entropy:** The proposed MMI framework maximizes the sum of entropy and variational conditional probability, which provides a lower bound of mutual information between actions. As aforementioned, maximizing the entropy and variational conditional probability enhance exploration and predictability for other agents' actions, respectively. Hence, the proposed MMI framework enhances correlated exploration among agents. We compared VM3-AC with MA-SAC, which is an extension of maximum entropy RL to multi-agent setting. We performed an experiment in the predator-prey environment with four agents where the number of required agents to catch the prey is two. In this environment, the agents started at the center of the map. Hence, the agents should spread out in the group of two to catch preys efficiently. Fig.4 shows the positions of the four agents at five time-steps after the episode starts. The first row and the second row in Fig.4 show the results of VM3-AC and MA-SAC in the early stage of the training, respectively. It is seen that the agents of VM3-AC explore in the group of two while the agents of MA-SAC tend to explore independently. We provided the performance comparisons of VM3-AC with MA-SAC in Fig.5 (a) and (b).

**Latent variable:** The role of the latent variable is to induce mutual information among actions and promote coordinated behavior. We compared VM3-AC and VM3-AC without the latent variable (implemented by setting $\dim(z) = 0$) in the multi-walker environment. In both cases, VM3-AC yields better performance that VM3-AC without the latent variable as shown in Fig.5(a) and 5(b).

**Injecting zero vector instead of the latent variable:** As mentioned in Section 5, we replace the latent variable with zero vector to execute actions without communication in the execution phase. We compared the performance of decentralized policies which use zero vector and decentralized policies which use the latent variable assuming communication. We used deterministic evaluation based on 20 episodes generated by the corresponding deterministic policy, i.e., each agent selects action using the mean network of Gaussian policy $\pi^i_{\phi^i}$. We averaged the return over 7 seeds, and the result

Table 1: Impact of replacing the latent variable $z \sim \mathcal{N}(0, I)$ with zero vector $z = \overrightarrow{0}$ in the execution phase

|  | PP (N=2) | PP (N=3) | PP (N=4) |
|---|---|---|---|
| $z \sim \mathcal{N}(0, I)$ | 413 | 734 | 1123 |
| $z = \overrightarrow{0}$ | 409 | 743 | 1147 |

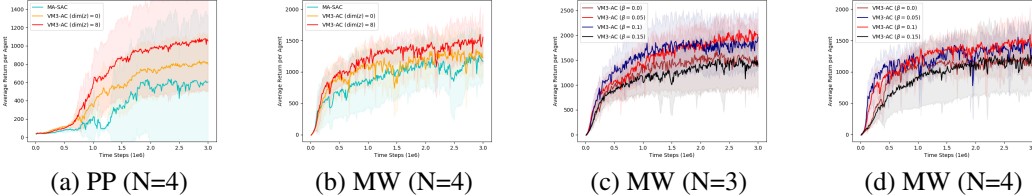

| (a) PP (N=4) | (b) MW (N=4) | (c) MW (N=3) | (d) MW (N=4) |

Figure 5: (a) and (b): VM3-AC (red), VM3-AC without latent variable (orange), and MA-SAC (cyan) and (c) and (d): performance with respect to the temperature parameter

is shown in Table 3. It is seen that the zero vector replacement method yields almost the same performance and enables fully decentralized execution without performance loss.

**Temperature parameter $\beta$:** The role of temperature parameter $\beta$ is to control the relative importance between the reward and the mutual information. We evaluated VM3-AC by varying $\beta = [0, 0.05, 0.1, 0.15]$ in the multi-walker environment with $N = 3$ and $N = 4$. Fig. 5(c) and 5(d) show that VM3-AC with the temperature value around $[0.05, 0.1]$ yields good performance.

## 7 CONCLUSION

In this paper, we have proposed the MMI framework for MARL to enhance multi-agent coordinated learning under CTDE by regularizing the cumulative return with mutual information among actions. The MMI framework is implemented practically by using a latent variable and variational technique and applying approximate policy iteration. Numerical results show that the derived algorithm named VM3-AC outperforms other baselines, especially in multi-agent tasks requiring high coordination among agents. Furthermore, the MMI framework can be combined with other techniques for cooperative MARL, such as value decomposition (Rashid et al. (2018)) to yield better performance.

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

APPENDIX A: VARIATIONAL POLICY EVALUATION AND POLICY IMPROVEMENT

In the main paper, we defined the state and state-action value functions for each agent as follows:

$$V_i^{\boldsymbol{\pi}}(s) \triangleq E_{\boldsymbol{\pi}} \left[ \sum_{t=0}^{\infty} \gamma^t \left( r_t + \beta H(\pi^i(\cdot|s_t)) + \frac{\beta}{N} \sum_{j \neq i} \log q^{(i,j)}(a_t^i, a_t^j, s_t) \right) \middle| s_0 = s \right] \quad (14)$$

$$Q_i^{\boldsymbol{\pi}}(s,a) \triangleq E_{\boldsymbol{\pi}} \left[ r_0 + \gamma V_i^{\boldsymbol{\pi}}(s_1) \middle| s_0 = s, a_0 = a \right], \quad (15)$$

**Lemma 3.** *(Variational Policy Evaluation). For fixed $\boldsymbol{\pi}$ and the variational distribution $q$, consider the modified Bellman operator $\mathcal{T}^{\boldsymbol{\pi}}$ in (16) and an arbitrary initial function $Q_i^0 : \mathcal{S} \times \mathcal{A} \to \mathbb{R}$, and define $Q_i^{k+1} = \mathcal{T}^{\boldsymbol{\pi}} Q_i^k$. Then, $Q_i^k$ converges to $Q_i^{\boldsymbol{\pi}}$ defined in (15).*

$$\mathcal{T}^{\boldsymbol{\pi}} Q_i(s, \boldsymbol{a}) \triangleq r(s, \boldsymbol{a}) + \gamma E_{s' \sim p}[V_i(s')], \quad (16)$$

where

$$V_i(s) = E_{\boldsymbol{a} \sim \boldsymbol{\pi}} \left[ Q_i(s, \boldsymbol{a}) - \beta \log \pi^i(a^i|s) + \frac{\beta}{N} \sum_{j \neq i} \log q^{(i,j)}(a^i, a^j, s) \right] \quad (17)$$

*Proof.* Define the mutual information augmented reward as $\mathcal{T}^{\pi} Q_i(s_t, \boldsymbol{a_t}) =$

$$= r(s_t, \boldsymbol{a_t}) + \gamma E_{s_{t+1} \sim p, \boldsymbol{a_{t+1}} \sim \boldsymbol{\pi}} \left[ Q_i(s_{t+1}, \boldsymbol{a_{t+1}}) - \beta \log \pi^i(a_t^i|s_t) + \frac{\beta}{N} \sum_{j \neq i} \log q^{(i,j)}(a_t^i, a_t^j, s_t) \right]$$

$$(18)$$

$$= r(s_t, \boldsymbol{a_t}) + \gamma E_{s_{t+1} \sim p, \boldsymbol{a_{t+1}} \sim \boldsymbol{\pi}} \underbrace{\left[ -\beta \log \pi^i(a_t^i|s_t) + \frac{\beta}{N} \sum_{j \neq i} \log q^{(i,j)}(a_t^i, a_t^j, s_t) \right]}_{r_\pi(s_t, \boldsymbol{a_t})} \quad (19)$$

$$+ \gamma E_{s_{t+1} \sim p, \boldsymbol{a_{t+1}} \sim \boldsymbol{\pi}} \left[ Q_i(s_{t+1}, \boldsymbol{a_{t+1}}) \right] \quad (20)$$

$$= r_\pi(s_t, \boldsymbol{a_t}) + \gamma E_{s_{t+1} \sim p, \boldsymbol{a_{t+1}} \sim \boldsymbol{\pi}} \left[ Q_i(s_{t+1}, \boldsymbol{a_{t+1}}) \right] \quad (21)$$

Then, we can apply the standard convergence results for policy evaluation. Define

$$\mathcal{T}^\pi(v) = \mathcal{R}^\pi + \gamma \mathcal{P}^\pi v \quad (22)$$

for $v = [Q(s, \boldsymbol{a})]_{s \in \mathcal{S}, \boldsymbol{a} \in \mathcal{A}}$. Then, the operator $\mathcal{T}^\pi$ is a $\gamma$-contraction.

$$\|\mathcal{T}^\pi(v) - \mathcal{T}^\pi(u)\|_\infty = \|(\mathcal{R}^\pi + \gamma \mathcal{P}^\pi v) - (\mathcal{R}^\pi + \gamma \mathcal{P}^\pi u)\|_\infty \quad (23)$$

$$= \|\gamma \mathcal{P}^\pi(v - u)\|_\infty \quad (24)$$

$$\leq \|\gamma \mathcal{P}^\pi\|_\infty \|v - u\|_\infty \quad (25)$$

$$\leq \gamma \|u - v\|_\infty \quad (26)$$

Note that the operator $\mathcal{T}^\pi$ has an unique fixed point by the contraction mapping theorem, and we define the fixed point as $Q_i^\pi(s, \boldsymbol{a})$. Since

$$\|Q_i^k(s, \boldsymbol{a}) - Q_i^\pi(s, \boldsymbol{a})\|_\infty \leq \gamma \|Q_i^{k-1}(s, \boldsymbol{a}) - Q_i^\pi(s, \boldsymbol{a})\|_\infty \leq \cdots \leq \gamma^k \|Q_i^0(s, \boldsymbol{a}) - Q_i^\pi(s, \boldsymbol{a})\|_\infty, \quad (27)$$

we have

$$\lim_{k \to \infty} \|Q_i^k(s, \boldsymbol{a}) - Q_i^\pi(s, \boldsymbol{a})\|_\infty = 0 \quad (28)$$

and this implies

$$\lim_{k \to \infty} Q_i^k(s, \boldsymbol{a}) = Q_i^\pi(s, \boldsymbol{a}), \quad \forall (s, \boldsymbol{a}) \in (\mathcal{S} \times \boldsymbol{\mathcal{A}}). \tag{29}$$

□

We proved the variational policy evaluation in the finite state-action sets. We can expand it to the infinite state-action sets by assuming follows:

- Assume that Q functions for $\pi$ are in L infinity

- From Folland (1999), L infinity is Banach space

- From Agarwal et al. (2018), by Banach fixed point theorem, Q function should be converge to a unique point in L infinity space and that is the Q function of given $\pi$

**Lemma 4.** *(Variational Policy Improvement). Let $\pi_{new}^i$ and $q_{new}$ be the updated policy and the variational distribution from (30). Then, $Q_i^{\pi_{new}^i, \pi_{old}^{-i}}(s, \boldsymbol{a}) \geq Q_i^{\pi_{old}^i, \pi_{old}^{-i}}(s, \boldsymbol{a})$ for all $(s, \boldsymbol{a}) \in (\mathcal{S} \times \boldsymbol{\mathcal{A}})$.*

$$(\pi_{k+1}^i, q_{k+1}) = \arg\max_{\pi^i, q} E_{(a^i, a^{-i}) \sim (\pi^i, \pi_k^{-i})} \left[ Q_i^{\boldsymbol{\pi}_k}(s, \boldsymbol{a}) - \beta \log \pi^i(a^i|s) \right. \tag{30}$$

$$\left. + \frac{\beta}{N} \sum_{j \neq i} \log q^{(i,j)}(a^i, a^j, s)) \right], \tag{31}$$

*Proof.* Let $\pi_{new}$ be determined as

$$(\pi_{new}^i, q_{new}) = \arg\max_{\pi^i, q} E_{(a_t^i, a_t^{-i}) \sim (\pi^i, \pi_{old}^{-i})} \left[ Q_i^{\boldsymbol{\pi}_{old}}(s_t, \boldsymbol{a}_t) - \beta \log \pi^i(a_t^i|s_t) \right. \tag{32}$$

$$\left. + \frac{\beta}{N} \sum_{j \neq i} \log q^{(i,j)}(a_t^i, a_t^j, s_t)) \right]. \tag{33}$$

Then, the following inequality is hold

$$E_{(a_t^i, a_t^{-i}) \sim (\pi_{new}^i, \pi_{old}^{-i})} \left[ Q_i^{\boldsymbol{\pi}_{old}}(s_t, \boldsymbol{a}_t) - \beta \log \pi_{new}^i(a_t^i|s_t) + \frac{\beta}{N} \sum_{j \neq i} \log q_{new}^{(i,j)}(a_t^i, a_t^j, s_t)) \right] \tag{34}$$

$$\geq E_{(a_t^i, a_t^{-i}) \sim (\pi_{old}^i, \pi_{old}^{-i})} \left[ Q_i^{\boldsymbol{\pi}_{old}}(s_t, \boldsymbol{a}_t) - \beta \log \pi_{old}^i(a_t^i|s_t) + \frac{\beta}{N} \sum_{j \neq i} \log q_{old}^{(i,j)}(a_t^i, a_t^j, s_t)) \right] \tag{35}$$

$$= V_i^{\boldsymbol{\pi}_{old}}(s_t). \tag{36}$$

From the definition of the Bellman operator,

$$Q_i^{\boldsymbol{\pi}_{old}}(s_t, \boldsymbol{a_t}) = r(s_t, \boldsymbol{a_t}) + \gamma E_{s_{t+1} \sim p}[V_i^{\boldsymbol{\pi}_{old}}(s_{t+1})] \tag{37}$$

$$\leq r(s_t, \boldsymbol{a_t}) + \gamma E_{s_{t+1} \sim p} E_{(a_{t+1}^i, a_{t+1}^{-i}) \sim (\pi_{new}^i, \pi_{old}^{-i})} \Bigg[ Q_i^{\boldsymbol{\pi}_{old}}(s_{t+1}, \boldsymbol{a_{t+1}})$$

$$- \beta \log \pi_{new}^i(a_{t+1}^i | s_{t+1}) + \beta \sum_{j \neq i} \log q_{new}^{(i,j)}(a_{t+1}^i, a_{t+1}^j, s_{t+1}) \Bigg] \tag{38}$$

$$\leq r(s_t, \boldsymbol{a_t}) + \gamma E_{s_{t+1} \sim p} E_{(a_{t+1}^i, a_{t+1}^{-i}) \sim (\pi_{new}^i, \pi_{old}^{-i})} \Bigg[ r^i(s_{t+1}, \boldsymbol{a_{t+1}})$$

$$- \beta \log \pi_{new}^i(a_{t+1}^i | s_{t+1}) + \beta \sum_{j \neq i} \log q_{new}^{(i,j)}(a_{t+1}^i, a_{t+1}^j, s_{t+1}) + \gamma V_i^{\boldsymbol{\pi}_{old}}(s_{t+2}) \Bigg]$$

$$\tag{39}$$

$$\vdots$$

$$\leq Q_i^{\pi_{new}^i, \pi_{old}^{-i}}(s_t, a_t). \tag{40}$$

$\square$

## APPENDIX B: ALGORITHM CONSTRUCTION

### 7.1 CENTRALIZED TRAINING

The value functions $V_{\psi_i}^i(\boldsymbol{x})$, $Q_{\theta_i}^i(\boldsymbol{x}, \boldsymbol{a})$ are updated based on the modified Bellman operator defined in (16) and (17). The state-value function $V_{\psi_i}^i(\boldsymbol{x})$ is trained to minimize the following loss function:

$$\mathcal{L}_V(\psi^i) = E_{s_t \sim D}\left[\frac{1}{2}(V_{\psi^i}^i(\boldsymbol{x}_t) - \hat{V}_{\psi^i}^i(\boldsymbol{x}_t))^2\right] \tag{41}$$

where $\quad \hat{V}_{\psi^i}^i(\boldsymbol{x}_t) \quad = \quad E_{z \sim N(0,I), \{a^i \sim \pi^i(\cdot|o_t^i, z)\}_{i=1}^N}\left[Q_{min}^i(\boldsymbol{x}_t, \boldsymbol{a}_t) \quad - \quad \beta \log \pi_{\phi^i}^i(a_t^i|o_t^i) \quad +$

$\frac{\beta}{N}\sum_{j \neq i} \log q_{\xi^i}^{(i,j)}(a_t^i, a_t^j, o_t^i, o_t^j)\Big]$, $D$ is the replay buffer that stores the transitions $(\boldsymbol{x}_t, \boldsymbol{a}_t, r_t, \boldsymbol{x}_{t+1})$,

and $Q_{min}^i(\boldsymbol{x}_t, a_t^i) = \min[Q_{\theta^{i,1}}^i(\boldsymbol{x}_t, a_t^i), Q_{\theta^{i,2}}^i(\boldsymbol{x}_t, a_t^i)]$ is the minimum of the two action-value functions to prevent the overestimation problem Fujimoto et al. (2018). The two action-value functions are updated by minimizing the loss

$$\mathcal{L}_Q(\theta^i) = E_{(\boldsymbol{x}_t, \boldsymbol{a}_t) \sim D}\left[\frac{1}{2}(Q_{\theta^i}(\boldsymbol{x}_t, \boldsymbol{a}_t) - \hat{Q}(\boldsymbol{x}_t, \boldsymbol{a}_t))^2\right] \tag{42}$$

where

$$\hat{Q}(\boldsymbol{x}_t, \boldsymbol{a}_t) = r_t(x_t, \boldsymbol{a_t}) + \gamma E_{\boldsymbol{x}_{t+1}}[V_{\overline{\psi}^i}(x_{t+1})] \tag{43}$$

and $V_{\overline{\psi}^i}$ is the target value network, which is updated by the exponential moving average method. We implement the reparameterization trick to estimate the stochastic gradient of policy loss. Then, the action of agent $i$ is given by $a^i = f_{\phi^i}(s; \epsilon^i, z)$, where $\epsilon^i \sim \mathcal{N}(0, I)$ and $z \sim \mathcal{N}(0, I)$. The policy for agent $i$ and the variational distribution are trained to minimize the following policy improvement loss,

$$\mathcal{L}_{\pi^i, q}(\phi^i, \xi) = E_{\substack{s_t \sim D, \\ \epsilon^i \sim \mathcal{N}, \\ z \sim \mathcal{N}}}\left[-Q_{\theta^{i,1}}^i(\boldsymbol{x}_t, \boldsymbol{a}) + \beta \log \pi_{\phi^i}^i(a^i|o_t^i) - \frac{\beta}{N}\sum_{j \neq i} \log q_{\xi^i}^{(i,j)}(a^i, a^j, o_t^i, o_t^j)\right] \tag{44}$$

where

$$q_{\xi^i}^{(i,j)}(a_t^i, a_t^j, o_t^i, o_t^j) = \underbrace{q_{\xi^i}(a_t^i|a_t^j, o_t^i, o_t^j)}_{(a)} \underbrace{q_{\xi^i}(a_t^j|a_t^i, o_t^i, o_t^j)}_{(b)}. \tag{45}$$

Since approximation of the variational distribution is not accurate in the early stage of training and the learning via the term (a) in equation 45 is more susceptible to approximation error, we propagate the gradient only through the term (b) in equation 45 to make learning stable. Note that minimizing $-\log q_{\xi^i}(a^j|a^i, s_t)$ is equivalent to minimizing the mean-squared error between $a^j$ and $\mu_{\xi^i}(a^i, o^i, o^j)$ due to our Gaussian assumption on the variational distribution.

APPENDIX C: PSEUDO CODE

---

**Algorithm 1** VM3-AC (L=1)

---

**Centralized training phase**

Initialize parameter $\phi^i, \theta^i, \psi^i, \overline{\psi}^i, \xi^i, \forall i \in \{1, \cdots, N\}$

**for** $episode = 1, 2, \cdots$ **do**

    Initialize state $s_0$ and each agent observes $o_0^i$

    **for** $t < T$ and $s_t \neq$ terminal **do**

        Generate $z_t \sim \mathcal{N}(0, I)$ and select action $a_t^i \sim \pi^i(\cdot | o_t^i, z_t)$ for each agent $i$

        Execute $\boldsymbol{a_t}$ and each agent $i$ receives $r_t$ and $o_{t+1}^i$

        Store transitions in $D$

    **end for**

    **for** each gradient step **do**

        Sample a minibatch from D and generate $z_l \sim \mathcal{N}(0, I)$ for each transition.

        Update $\theta^i, \psi^i$ by minimizing the loss (42) and (43)

        Update $\phi^i, \xi^i$ by minimizing the loss (44)

    **end for**

    Update $\overline{\psi}^i$ using the moving average method

**end for**

**Decentralized execution phase**

Initialize state $s_0$ and each agent observes $o_0^i$

**for** each environment step **do**

    Select action $a_t^i \sim \pi^i(\cdot | o_t^i, z_t)$ where $z_t = \overrightarrow{0}$ (or sample from the Gaussian random sequence generator with the same seed)

    Execute $\boldsymbol{a_t}$ and each agent $i$ receives $o_{t+1}^i$

**end for**

---

APPENDIX D: ENVIRONMENT DETAIL

**Multi-walker** The multi-walker environment, which was introduced in Gupta et al. (2017), is a modified version of the BipedalWalker environment in OpenAI gym to multi-agent setting. The environment consists of $N$ bipedal walkers and a large package. The goal of the environment is to move forward together while holding the large package on top of the walkers. The observation of each agent consists of the joint angular speed, the position of joints and so on. Each agent has 4-dimensional continuous actions that control the torque of their legs. Each agent receives shared reward $R_1$ depending on the distance over which the package has moved and receives negative local compensation $R_2$ if the agent drops the package or falls to the ground. An episode ends when one of the agents falls, the package is dropped or $T$ time steps elapse. To obtain higher rewards, the agents should learn coordinated behavior. For example, if one agent only tries to learn to move forward, ignoring other agents, then other agents may fall. In addition, the different coordinated behavior is required as the number of agents changes. We set $T = 500$, $R_2 = -10$ and $R_1 = 10d$, where $d$ is the distance over which the package has moved. We simulated this environment in three cases by changing the number of agents ($N = 2$, $N = 3$, and $N = 4$).

All algorithms used neural networks to approximate the required functions. In the algorithms except I-SAC, we used the neural network architecture proposed in Kim et al. (2019) to emphasize the agent's own observation and action for centralized critics. For agent $i$, we used the shared neural network for the variational distribution $q_{\xi^i}(a_t^j | a_t^i, o_t^i, o_t^j)$ for $j \in \{1, \cdots, N\} \setminus \{i\}$, and the network takes the one-hot vector which indicates $j$ as input. Experimental details are given in Appendix E.

**Predator-prey** The predator-prey environment, which is a standard task for MARL, consists of $N$ predators and $M$ preys. We used a variant of the predator-prey environment into the continuous domain. The initial positions on the predators are randomly determined, and those of the preys are in the shape of a square lattice as shown in figure6 (b). The goal of the environment is to capture as many preys as possible during a given time $T$. A prey is captured when $C$ predators catch the prey simultaneously. The predators get team reward $R_1$ when they catch a prey. After all of the preys are captured and removed, we set the preys to respawn

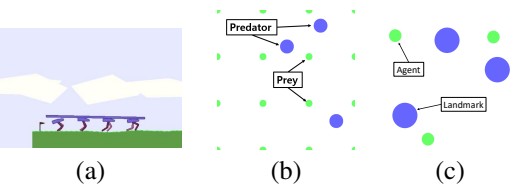

(a)  (b)  (c)

Figure 6: Considered environments: (a) Multi-walker, (b) Predator-prey, and (c) Cooperative navigation

in the same position and increase the value of $R_1$. Thus, the different coordinated behavior is needed as $N$ and $C$ change. The observation of each agent consists of relative positions between agents and other agents and those between agents and the preys. Thus, each agent can access to all information of the environment state. The action of each agent is two-dimensional physical action. We set $R_1 = 10$ and $T = 100$. We simulated the environment with three cases: ($N = 2, M = 16, C = 1$), ($N = 3, M = 16, C = 1$) and ($N = 4, M = 16, C = 2$).

**Cooperative navigation** Cooperative navigation, which was proposed in Lowe et al. (2017), consists of $N$ agents and $L$ landmarks. The goal of this environment is to occupy all landmarks while avoiding collision with other agents. The agent receives shared reward $R_1$ which is the sum of the minimum distance of the landmarks from any agents, and the agents who collide each other receive negative reward $-R_2$. In addition, all agents receive $R_3$ if all landmarks are occupied. The observation of each agent consists of the locations of all other agents and landmarks, and action is two-dimensional physical action. We set $R_2 = 10$, $R_3 = 1$, and $T = 50$. We simulated the environment in the cases of ($N = 3, L = 3$).

APPENDIX E: HYPERPARAMETER AND TRAINING DETAIL

The hyperparameters for MA-AC, MA-SAC, MADDPG, and VM3-AC are summarized in Table 2.

Table 2: Hyperparameters of all algorithms

|  | MA-AC | MA-SAC | MADDPG | VM3-AC |
|---|---|---|---|---|
| REPLAY BUFFER SIZE | $5 \times 10^5$ | $5 \times 10^5$ | $5 \times 10^5$ | $5 \times 10^5$ |
| DISCOUNT FACTOR | 0.99 | 0.99 | 0.99 | 0.99 |
| MINI-BATCH SIZE | 128 | 128 | 128 | 128 |
| OPTIMIZER | ADAM | ADAM | ADAM | ADAM |
| LEARNING RATE | 0.0003 | 0.0003 | 0.0003 | 0.0003 |
| TARGET SMOOTHING COEFFICIENT | 0.005 | 0.005 | 0.005 | 0.005 |
| NUMBER OF HIDDEN LAYERS (ALL NETWORKS) | 2 | 2 | 2 | 2 |
| NUMBER OF HIDDEN UNITS PER LAYER | 128 | 128 | 128 | 128 |
| ACTIVATION FUNCTION FOR HIDDEN LAYER | ReLU | ReLU | ReLU | ReLU |
| ACTIVATION FUNCTION FOR FINAL LAYER | TANH | TANH | TANH | TANH |

Table 3: The temperature parameter $\beta$ and the dimension of the latent variable $z$ for VM3-AC on the considered environments. Note that the temperature parameter $\beta$ in I-SAC and MA-SAC controls the relative importance between the reward and the entropy, whereas the temperature parameter $\beta$ in VM3-AC controls the relative importance between the reward and the mutual information.

| VM3-AC | $\beta$ | DIM(Z) |
|---|---|---|
| MW (N=3) | 0.05 | 8 |
| MW (N=4) | 0.1 | 8 |
| PP (N=2) | 0.15 | 8 |
| PP (N=3) | 0.1 | 8 |
| PP (N=4) | 0.2 | 8 |
| CN (N=3) | 0.1 | 8 |

