# OpenReview forum: "A Maximum Mutual Information Framework for Multi-Agent Reinforcement Learning"
_ICLR.cc/2021/Conference — Reject_

### Official Review · AnonReviewer3 · 2020-10-28
**The contribution of this paper largely overlaps with previous works.**

**Rating:** 3
**Confidence:** 4

**Review:**

This paper proposes regularizing the conventional MARL learning objective with a mutual information term to encourage more correlated behaviors among different agents. The contribution is clearly stated. However, the similarity to previous works is not sufficiently discussed, and the paper leaves out some important related works.

The paper maximizes the mutual information between agents' policies, given the current states. To allow policies to be conditional dependent, the authors assume there exists a dummy variable. The contributions of the paper about the lower bound of the mutual information (Sec. 4.2) and policy update (Sec. 4.3) are not significant. The lower bound of the mutual information has recently been extensively explored in multi-agent settings, used for encouraging role emergence, minimized communication, and exploration. The policy update and policy improvement guarantee can be easily obtained based on soft reinforcement learning literature.

** Major concern: About related works. My major concern is not about the two contributions mentioned above. Instead, I think that the first and main contribution of this paper is a subset of previous works' contributions. EDTI [1] discusses how to maximize the mutual information between the \emph{trajectories} of different agents. Their discussion already covers the correlation of policies of different agents at \emph{a single timestep}. More importantly, the authors of that paper also point out that only optimizing mutual information between trajectories is not enough because the reward signal has to be considered for better policy learning. They even discuss the second-order influence between these mutual-information-based intrinsic rewards. The contribution of this paper seems to be the first part of [1]. Frans Oliehoek and other researchers also did lots of excellent works on this topic [2,3,4,5]. However, the discussion about these related works is absent from this paper.

Additionally, the difference between the proposed method and Jaques's paper (social influence) is not significant. The only difference is whether the action of other agents, $a_j$, is $a_j^{t}$ or $a_j^{t+1}$ when calculating the mutual information (In Jaques's paper, they prove that their formulation is equivalent to a mutual information formulation). I do not think the author's definition is an improvement of that of Jaques. At least, the authors should provide a more serious discussion about this point, perhaps providing a matrix game to show that different timesteps do indeed make a difference. The authors may argue that their experiments show that their method has better performance. The problem is that SSD used by Jaques is a more challenging task than those used in this paper. Moreover, social influence is sensitive to hyperparameter settings and needs fine-tuning to reach its full potential. I will change my mind regarding the experiments if the authors could provide SSD results (adopting their methods to tasks with discrete actions is not very difficult). Besides, Jaques et al. also discuss how to make condition dependency clear between agents and how to carry out influential communications, which are not discussed in this paper.

[1] Wang, T., Wang, J., Wu, Y. and Zhang, C., 2019. Influence-based multi-agent exploration. ICLR 2020 spotlight

[2] F. A. Oliehoek, S. Witwicki, and L. P. Kaelbling. Influence-based abstraction for multiagent systems. In Proceedings of the Twenty-Sixth AAAI Conference on Artificial Intelligence, pages 1422–1428, July 2012
* Also see a recent longer version: https://arxiv.org/abs/1907.09278

[3] R. Becker, S. Zilberstein, V. Lesser, and C. V. Goldman. Transition-independent decentralized Markov decision processes. In Proceedings of the International Conference on Autonomous Agents and Multiagent Systems, pages 41–48, 2003.

[4] Miguel Suau de Castro, Elena Congeduti, Rolf A.N. Starre, Aleksander Czechowski, and Frans A. Oliehoek. Influence-Based Abstraction in Deep Reinforcement Learning. In Proceedings of the AAMAS Workshop on Adaptive Learning Agents (ALA), May 2019.

[5] Frans A. Oliehoek and Christopher Amato. A Concise Introduction to Decentralized POMDPs, SpringerBriefs in Intelligent Systems, Springer, May 2016.

---

> ### Author Response · Authors · 2020-11-23
> **Response to Reviewer3**
>
> Thank you for the valuable comments. Here is our reply:
>
> Regarding the related works:
> - We disagree with the reviewer on that the main contribution of our paper is a subset of [R-1]. We explained  the difference between our approach and [R-1] in the revised paper. [R-1] used the mutual information between (an agent's current actions and states) and (other agents' next states). That is, [R-1] considers the influence of an agent's current action on the next states of other agents. This work is in the line of social influence [R-2].
>     However, our work is not based on the influence but based on correlation between the agents' simultaneous actions by considering the mutual information between current actions of the agents.
>     In addition, [R-1] assumes the independent policies with no correlation, whereas our approach considers correlated policies by introducing a latent variable. This is a clear distinction.
>
>
>  - As mentioned in our paper, [R-2] uses the  mutual information $I(a_t^k; a_{t+1}^j| s_t)$ between influencer's action and influencee's next timestep action \textbf{not}
> the mutual information $I(a_t^k ; a_t^j| s_t)$ between agents' simultaneous actions. This is one of the main differences between our approach and [R-2]. In addition, our paper provides a theoretical framework for optimizing mutual information between agents' simultaneous actions by introducing a latent variable. Lastly, we have already provided comparison with the MOA method. It is shown that VM3-AC significantly outperforms the MOA method, and indeed coordinating simultaneous actions is more beneficial. Please see Fig. 3 for this comparison.  This performance improvement over [R-2] is one of our contributions in the current MARL area.
>
>
> [R-1] Wang, Tonghan, et al. "Influence-based multi-agent exploration." arXiv preprint arXiv:1910.05512 (2019).
>
> [R-2] Jaques, Natasha, et al. "Social influence as intrinsic motivation for multi-agent deep reinforcement learning." International Conference on Machine Learning. PMLR, 2019.

---

### Official Review · AnonReviewer4 · 2020-10-29
**An Interesting Mutual Information Based Learning Framework for Multiagent Cooperation**

**Rating:** 5
**Confidence:** 4

**Review:**

Summary:
- This paper proposes a Maximum Mutual Information framework for cooperative MARL. Following the insight that mutual information of agents’ policies is the indicator of coordination, this paper proposes VM3-AC, an MA-AC algorithm that optimizes long-term reward as well as a variational lower bound of mutual information in the paradigm of CTDE. Experimental results show superiority of proposed algorithm in comparison with a few benchmark approaches.


Detailed comments:
- I appreciate the idea of utilizing mutual information (MI) of agents’ policies to facilitate the coordination of cooperative multiple agents. In my personal opinion, the MI of agents’ policy can be viewed as an indicator of coordination, although the coordination may not be a good/optimal one (the results of temperature parameter experiments also demonstrate this. I will mention this point later below).
- My most concern is about the latent variable $z$. I agree that the latent variable $z$ should be an unobserved variable which reflects some information of coordination, e.g., such a variable can be implicitly induced during the learning process of policies of agents. In this paper, the latent variable $z$ in both learning and execution. Especially, the authors use a common and pre-determined sequence of $z$ is generated before execution for agents. In my opinion, this violates the CTDE paradigm which is claimed by the authors, since such a common sequence of $z$ is more like explicit signals of how to perform coordination (not truly decentralized). I am willing to hear the further understanding about the practical role of $z$ from the authors.
- Moreover, it seems that the two of three environments, i.e., Predator-prey and Cooperative Navigation, are not partially observable (PO). It would be better to provide more evaluation under PO environments.


Questions:
- Can the authors further discuss the reason to stop the gradient of term (a) in Equation 45, since such a mechanism results in a practical optimization objective different from variation approximate MI.
- As in Algorithm 1, in decentralized execution phase, the latent $z$ is set to zero-vector, which conflicts the random sequence generated from the same random process as described in Section 5. Which one is the practical way and what is the difference in learning performance?
- Below Equation 13, it says that $L = 1$ is used for MC expectation. Is $L = 1$ is sufficient for good performance? And how can larger values of $L$ influence the learning process?

Suggestions:
- As mentioned above, I view MI as an indicator of coordination, however, the coordination may not be preferred. Therefore, maximize the MI sometimes may trap the agents in a sub-optimal coordination (although the reward-objective may help them out to some degree). One potential evidence is, a larger temperature parameter (0.15) hampers the learning performance, as shown in Figure 5 (c)-(d). Besides, the stabilization induced by stop partial gradients of Equation 45 may also be explained by this point. Therefore, a more sophisticated mechanism for adaptive optimization of MI can be considered.
- One possible concern (also possible future work) of proposed approach is the current modeling of MI is pair-wise modeling, of which the apparent drawback is the computational complexity can increase quadratically to the agent number. As in this paper, the experiments contain environments with up to 4 agents. A development towards more agents should be considered.


Minors:
- In Figure 5, last two sub-plots are both labeled as (c). Temperature parameters are denoted by \alpha in legends but \beta in paragraph.


Overall, I think this paper takes a good attempt to study MI in cooperative MARL, proposing a reasonable algorithm with promising experimental results.

---

> ### Author Response · Authors · 2020-11-23
> **Response to Reviewer3**
>
>
> Thank you for the valuable comments. Here is our reply:
>
> Regarding the latent variable:
> - As mentioned in the decentralized execution paragraph in Section 5, sharing the latent variable can be done by making all agents have the same Gaussian random sequence generator and distributing the same seed to this random sequence generator only once in the beginning of the
> execution phase. Please note that such random sequence generators are used in many places such randn() in C. So, distributing the seed only does not seem to be a major issue.
> In addition, we can consider an alternative method to  achieve decentralized execution. That is,  we simply replace $z^l$ with
>  zero vector with matching dimensions in the decentralized execution phase.
>   This substitution method does not deteriorate the performance much as seen in Table 1 since the latent variable distribution is zero-mean Gaussian, and zero has the highest density.
>
>
>
>
> Regarding "the reason to stop the gradient of term (a) in Equation 45"
> -  The variational distribution is Gaussian distribution with constant variance and the mean is parameterized by a neural network. Thus maximizing term (a) in Equation 45, $\log q_{\xi^i}(a_t^i|a_t^j,o_t^i,o_t^j)$, is equal to minimizing $\| q_{\xi^i}(a_t^j,o_t^i,o_t^j) - {a_t^i}\|^2$, whereas maximizing term (b)  in Equation 45, $q_{\xi^i}(a_t^j|a_t^i,o_t^i,o_t^j)$, is equal to minimizing $\| q_{\xi^i}({a_t^i},o_t^i,o_t^j) - a_t^j\|^2$. Now let us consider the perspective of Agent $i$. In the early stage of learning, $q_{\xi^i}(a_t^j,o_t^i,o_t^j)$ has large approximation error, then the policy which generates $a_t^i$ can be unstable because the policy is trained to follow $q_{\xi^i}(a_t^j,o_t^i,o_t^j)$ with large noise.
>
>
>
> Regarding the MC expectation:
> - In our experiment, we observe that  $L=1$ is enough. The larger $L$ requires more computation.
>
>
>
>
>
> Regarding "the coordination may not be a good/optimal one" :
> - We agree on that coordination is not always needed. Some environments do not require coordination or agents do not need coordination every time step. In such situation, the policies of agents will be trained to be independent of each other, and then the mutual information between policies becomes zero. Thus, we think that the proposed regularization term, which is a lower bound on the mutual information, does not deteriorate the performance.
>
>
>
>
> Regarding the complexity:
> - We agree on that complexity becomes an issue as the number of agents increases. We can share the variational distribution among agents by injecting the one-hot vector indicating the agent index.

---

### Official Review · AnonReviewer1 · 2020-10-29
**Proposed technique seems effective and it is nicely motivated and well written**

**Rating:** 6
**Confidence:** 4

**Review:**

Summary:
The authors propose to include the mutual information between agents' simultaneous actions in the objective to encourage coordinated behaviour. To induce positive mutual information, the authors relax the assumption that the joint policy can be decomposed as the product of each agent's policy, independent of each other given the state, and they achieve so by introducing a latent variable that correlates agents behaviours. Since the mutual information is difficult to compute, the authors proposed to maximise a parametric lower bound. The algorithm is theoretically motivated as a policy iteration variation in its exact tabular form. But experiments are performed with neural network approximations on some environments. Numerical results show improvements over previous similar techniques.

Strong points:
The mutual information objective, the latent variable, the variational lower bound, and the algorithm are well motivated.
Paper is well written.
Simulation results seem convincing.

Weak points:
The decentralised execution relying on having random generators with the exact same seed is not a robust solution.
The presentation as modified policy iteration is OK, but quite straightforward. The characterisation as a contraction is similar to many other works, maybe a citation would have helped.


Questions:
Is always coordination desirable? Could be some adversarial example of an environment where the optimal policy requires lack of coordination? Wouldn't the current method suffer in such case?
Does the paper assume finite state-action sets? If so, please say it explicitly.
Since the environment has stochastic transitions, aren't more assumptions needed in order to allow gamma = 1 and still ensure existence of optimal policy?
Regarding the rightmost term of (b) in (7), is it missing from (6) and (8)?

Comments that didn't influence the score:
Last paragraph of page 2, "to explore widely" seems loose --> "to enhance exploration" might be more accurate.
The authors use the term causal diagram, but it seems to refer a Bayesian network, which indicates correlation rather than causality, is that right?
Third line of the conclusions paragraph in Sec. 7, wouldn't be more accurate to say "applying approximate policy iteration"?

---

> ### Author Response · Authors · 2020-11-23
> **Response to Reviewer 2**
>
> Thank you for the valuable comments. Here is our reply:
>
>
> Regarding "The decentralised execution relying on having random generators with the exact same seed is not a robust solution":
> - As mentioned in the decentralized execution paragraph in Section 5, sharing the latent variable can be done by making all agents have the same Gaussian random sequence generator and distributing the same seed to this random sequence generator only once in the beginning of the
> execution phase. Please note that such random sequence generators are used in many places such randn() in C. So, distributing the seed only does not seem to be a major issue.
> In addition, we can consider an alternative method to  achieve decentralized execution. That is,  we simply replace $z^l$ with
>  zero vector with matching dimensions in the decentralized execution phase. This substitution method does not deteriorate the performance much as seen in Table 1 since the latent variable distribution is zero-mean Gaussian, and zero has the highest density.
>
>
>
>
> Regarding "Is always coordination desirable?":
> - We agree on that coordination is not always needed. Some environments do not require coordination or agents do not need coordination every time step. In such situation, the policies of agents will be trained to be independent of each other, and then the mutual information between policies becomes zero. Thus, we think that the proposed regularization term, which is a lower bound on the mutual information, does not deteriorate the performance.
>
>
>
> Regarding "Does the paper assume finite state-action sets? If so, please say it explicitly.":
> - Our proof is based on finite state-action set, but the proof can be extended to the infinite state-action case by adding some assumptions. We added the necessary assumption in Appendix A.
>
>
>
>
> Regarding "Since the environment has stochastic transitions, aren't more assumptions needed in order to allow gamma = 1 and still ensure existence of optimal policy?"
> -We fixed the range of $\gamma$ to $[0,1)$ in the revised paper.
>
>
>
> Regarding "the rightmost term of (b) in (7), is it missing from (6) and (8)?":
> - Equations (6) and (8) include the the rightmost term of (b) in Equation (7). Please note that
> $\sum_{i=1}^{N}\sum_{j\neq i}\log q(a_t^j|a_t^i,s_t) = \sum_{j\neq i}\Big[\log q(a_t^i|a_t^j,s_t)+\log q(a_t^j|a_t^i,s_t)\Big]$. In addition, we define $q^{(i,j)}(a_t^i,a_t^j,s_t)\triangleq q(a_t^i|a_t^j,s_t)q(a_t^j|a_t^i,s_t)$ for equation (8).

---

> > ### Comment · AnonReviewer1 · 2020-11-23
> > **Further questions on some points**
> >
> > Thanks for the responses.
> >
> > The current proposal of having a common random generator doesn't requires high level of coordination typically achieved in a centralised manner, like deploying all the agents at the same time. However, in real deployments, agents are typically not deployed instantaneously, moreover they can fail and have to be replaced or new agents are added along time. Perhaps this could be surmounted with a synchronisation protocol (see e.g. the literature on wireless sensor networks). But I think this is definitely an issue from a practical point of view.
> >
> > I find difficult to understand how the mutual information term in the objective won't bias the solution to some coordination, even if the optimal solution is fully independent. I mean, once it is in the objective, the solution will find a tradeoff that maximises the sum of reward and regulariser. For example, if $\alpha \gg \sup{r_t(s_t, a_t)}$ and $\gamma \ll 1$, then I assume that the solution will be highly coordinated independent on the reward.

---

> > > ### Author Response · Authors · 2020-11-25
> > > **Response to Reviewer 2**
> > >
> > > Regarding "a practical point of view on having a common random generator":
> > >
> > > - We agree with the reviewer on the practical issue. We think that timestep synchronization may not be a big issue considering wireless sensor networks or the GPS technology. But, not simultaneous deployment may be an issue as the reviewer mentioned. One idea to circumvent this issue is as follows:  We simply set the latent variable as zero in the execution phase after training, although the latent variable in the training phase is actually random generated. Please note that 0 is the most probable sample in zero-mean Gaussian distribution. In this way, the agents will act in coordination corresponding to the case of the zero latent variable. It is observed that this method does not deteriorate the performance much, as seen in Table 1.
> > >
> > > Regarding "how the mutual information term in the objective won't bias the solution"
> > > - We agree on that "the solution will find a trade-off that maximizes the sum of reward and mutual information". Thus, choosing the temperature parameter (i.e., weighting factor) is important. One important fact is that with a proper temperature parameter choice, the pure reward performance by maximizing the sum of reward and mutual information is better than the reward performance by maximizing the reward alone,  as seen in Fig. 5 (c) and (d). We included an ablation study regarding the temperature parameter in section 6.2. This situation happens in many other RL algorithms.  One example is Soft Actor Critic (SAC) [R-3], which is the state-of-the-art RL algorithm in the continuous domain. SAC maximizes the sum of reward and the policy entropy, and outperforms the classical RL algorithms just maximizing the reward in terms of pure reward performance. This is because the additional regularization term enhances other aspects of the algorithm like exploration.   In our case, the added mutual information helps correlated exploration and this benefits the reward performance.
> > >
> > > [R-3] Haarnoja, Tuomas, et al. "Soft actor-critic: Off-policy maximum entropy deep reinforcement learning with a stochastic actor." arXiv preprint arXiv:1801.01290 (2018).

---

### Official Review · AnonReviewer2 · 2020-11-02
**More clarificaitons and experiments on MMI regularization**

**Rating:** 6
**Confidence:** 3

**Review:**

This paper can be seen as a modification of SAC, in a multiagent setup by adding the conditional entropy $H(\pi_i|\pi_j)$ as a second set of regularization on top of $H(\pi_i)$. The overall idea and intuition appear to be interesting.


1.The first question is whether the mutual information is informative enough. Mutual means two $a$, how about more, e.g., $H(a^i|a^j, a^k)$. When Eq.(2) is still our target, the underlying assumption is $a^j$ and $a^k$ would independently affect $a^i$. Is that true? And how the conditional distribution of $a^j$ on $a^k$ and vice versa would influence the objective?

2.The motivation for inducing mutual information using latent variable is not well motivated. I can only find the first sentence in Section 4.1.

3.Would the new regularization benefit other RL algorithms? Say, would the regularization combined with the baseline algorithms shown in the experiment be better compared with those without regularization?


4.The same latent variable $z$ is shared by all the policies. As shown in Fig. 1(b), would $z_{ij}$ make more sense? Essentially, would decentralized training be more reasonable?


5.In the experiment, multi-agent soft actor-critic is missing, i.e., only keep $H(\pi^j)$ in the regularization.

6.How is $\alpha$ selected?

---

> ### Author Response · Authors · 2020-11-23
> **Response to Reviewer 1**
>
> Thank you for your valuable comments. Here is our reply:
>
>
> 1.  Please note that $H(a^i|a^j)$ is derived from $I(a^i;a^j)$. As the reviewer mentioned, $H(a^i|a^j,a^k)$ can be helpful as our proposed method does. If we use the multivariate mutual information $I(a^1;\cdots;a^N)$ for the objective function instead of $\sum_{(i,j)}I(\pi^i(\cdot|s_t);\pi^j(\cdot|s_t)$, the term  $H(a^i|a^j,a^k)$ (mentioned by the reviewer) will appear. However, such an approach may be too complicated for practical implementation.
>
> 2.  The motivation for inducing mutual information using latent variable is described in the Section 4 (please see the paragraph after equation 2. "As aforementioned...."). In summary, under the assumption of decentralized learning (without a latent variable), the mutual information between policies is zero so that the problem simply reduces to conventional MARL.
>
>
> 3.  We expect that our proposed regularization can improve the performance of other MARL algorithms. As the entropy regularization is important to RL algorithms, the entropy and conditional entropy, which is a lower bound on mutual information, can be effective to MARL algorithms.
>
>
> 4.  As mentioned in the decentralized execution paragraph in Section 5, sharing the latent variable can be done by making all agents have the same Gaussian random sequence generator and distributing the same seed to this random sequence generator only once in the beginning of the
> execution phase. Please note that such random sequence generators are used in many places such randn() in C. So, distributing the seed only does not seem to be a major issue.
> In addition, we can consider an alternative method to  achieve decentralized execution. That is,  we simply replace $z^l$ with
>  zero vector with matching dimensions in the decentralized execution phase.
>   This substitution method does not deteriorate the performance much as seen in Table 1
>       since the latent variable distribution is zero-mean Gaussian, and zero has the highest density.
>
>
>
> 5.  We have already provided a result regarding comparison with MA-SAC, which only uses the entropy regularization, in section 6.2. Please see Fig. 5 (a) and (b).
>
>
>
> 6. $\alpha$ is a hyperparameter. We already included an ablation study regarding the temperature parameter in the section 6.2.  Please see Fig. 5 (c) and (d).

---

### Author Response · Authors · 2020-11-23
**Common Response**

Common response

We thank all reviewers for the valuable comments. We revised our paper based on the reviewers' comments.
- We added [R-1] into the related work section.
- We added an alternative method for sharing the latent variable in the decentralized execution phase and provided the results in the ablation study section. This is done by replacing $z^l$ with zero vector with the matching dimensions.


[R-1] Wang, Tonghan, et al. "Influence-based multi-agent exploration." arXiv preprint arXiv:1910.05512 (2019).

---

### Decision · Program_Chairs · 2021-01-07
**Final Decision**

**Decision:**

Reject

**Comment:**

Overview:
This paper introduces a maximum mutual information method for helping to coordinate RL agents without communication.

Discussion:
Some reviewers leaned towards accept, but I found the two reviewers recommending rejecting to be more convincing.

Recommendation:
This is an important research topic and I'm glad this paper is focusing on the problem. Hopefully the reviews will help improve a future version of this paper. I agree that this is a new way of using mutual information, but it seems more like a small improvement rather than a very significant step forward.

In addition, I think the setting needs to be better motivated. This is a centralized training with decentralized execution (CTDE) setting, and this paper helps the agents coordinate. In CTDE, the agents work in the environment and then pool their information to train before deploying on the next episode. I don't understand why, e.g., in multiwalker, agents would not be able to communicate while walking, can communicate after they succeed or drop the object (the episode ends), and then cannot communicate once the next episode starts.